# Enhancing Lignocellulose Degradation and Mycotoxin Reduction in Co-Composting with Bacterial Inoculation

**DOI:** 10.3390/microorganisms13030677

**Published:** 2025-03-18

**Authors:** Cheng Chen, Xiaolong Tang, Chaosheng Liao, Xiaokang Huang, Mingjie Zhang, Yubo Zhang, Pan Wang, Siqi Yang, Ping Li, Chao Chen

**Affiliations:** 1College of Animal Science, Guizhou University, Guiyang 550025, China; cc376651938@163.com (C.C.); txl971217@sina.com (X.T.); liaocs@yeah.net (C.L.); h15329727872@163.com (X.H.); 18286640532@163.com (M.Z.); zyb0129gz@sina.cn (Y.Z.); 18685204997@163.com (P.W.); 18685474242@163.com (S.Y.); lpyzm@sina.cn (P.L.); 2Key Laboratory of Animal Genetics, Breeding & Reproduction in the Plateau Mountainous Region, Ministry of Education, Guizhou University, Guiyang 550025, China

**Keywords:** mycotoxin-contaminated silage, mycotoxins, co-composting, bacterial agents

## Abstract

The burgeoning global silage industry has precipitated challenges related to the sustainable utilization of mycotoxin-contaminated silage. To understand the effect of bio-enhancement on lignocellulose degradation and mycotoxin reduction, mycotoxin-contaminated silage and rape straw were co-composted without (CK) or with different bacterial agents and their combinations. Compared to CK, the inoculation of *Weissella paramesenteroides* and *Bacillus subtilis* could increase the degradation rate of cellulose by 39.24% and lignin by 22.31% after composting. Inoculation of *W. paramesenteroides* and *Paenibacillus* sp. significantly enhanced cellulose and lignin degradation rates by 26.75% and 15.48%, respectively. Furthermore, this treatment significantly reduced mycotoxin levels (*p* < 0.05), including Aflatoxin B1 (AFB1, 64.48% reduction), T-2 toxin (65.02%), Ochratoxin A (OTA, 61.30%), Zearalenone (ZEN, 67.67%), and Vomitoxin (DON, 48.33%). Inoculation with *Paenibacillus* sp. and other bacteria increased total nitrogen by 48.34–65.52% through enhancing microbiological activity. Therefore, *Paenibacillus* sp. in combination with other bacteria could increase compost efficiency and reduce mycotoxin presence for better and safer utilization of agricultural waste by-products, enabling faster conversion of contaminated silage into safe soil amendments, which could reduce agricultural waste management costs.

## 1. Introduction

Composting is regarded as an efficient method for the ecological utilization of organic waste. It not only quickly converts organic waste into organic fertilizer but also helps to eliminate harmful substances such as ammonia and pathogens to a certain extent, thus protecting the ecological environment [1]. Oilseed rape, as one of the main oilseed crops, is an important source of edible oil in people’s daily lives [2]. With the continuous development of the rapeseed industry, the amount of rapeseed straw generated has also increased, making the proper disposal of rapeseed straw a major challenge [3,4]. The most primitive method of handling straw is burning it in the field, but this increases greenhouse gas emissions and exacerbates ecological climate conditions, making it the least desirable method currently. Composting is one of the green and ecological methods for managing straw, which not only increases the soil’s organic carbon reserves but reduces environmental pollution [5]. Rape straw has a high content of cellulose and hemicellulose, which is difficult to degrade, resulting in poor composting effectiveness [6]. Some studies have used the addition of manure and cellulose-degrading bacteria to utilize the microbial degradation capacity, shortening the degradation period of straw and retaining the nitrogen, phosphorus, and potassium components present in the straw.

Ensilage is a low-cost storage technique that allows fresh plants to be preserved for a long time, reducing nutrient loss and facilitating animal digestion and absorption [7]. A survey on the worldwide occurrence of mycotoxins revealed that 81% of 7049 livestock feed samples collected from the Americas, Europe, and Asia were positive for at least one mycotoxin [8]. If moldy silage is fed to livestock, it will cause harm to their bodies and transmit toxic substances into their products [9]. Moreover, due to the high levels of mycotoxins present in moldy silage, their disposal as ordinary waste can contaminate other plants or food [10]. The generation of such moldy silage causes losses for production enterprises and increases the cost of waste treatment. Therefore, finding a proper solution for the disposal of silage waste is a crucial challenge.

However, most research just focuses on the microbial cellulose degradation of straw composting and the enhancement of composting effectiveness by adding exogenous cellulose-degrading bacteria [11]. The existing literature on moldy silage composting is inadequate, and this study is uniquely innovative. It is also a harmless way to tackle moldy silage. However, competition between molds and cellulose-degrading bacteria during composting of moldy silage prolongs the composting period and affects material conversion. To promote material conversion, it is necessary to ensure a high abundance and activity of functional microorganisms [12]. Consequently, microbial agents composed of functional microorganisms are commonly used to improve composting efficiency and quality, as well as reduce environmental pollution [13]. Meanwhile, the excessive production of mycotoxins can affect composting quality and cause increased disease incidence and mycotoxin contamination in other plants or crops, such as tea plants being affected by fungi and mycotoxins in the soil during cultivation and harvesting [14]. Therefore, further research is needed on the effects of microbial agents on the co-composting of moldy silage and straw, including the relationships between composting components, mycotoxins, and bacterial communities. Additionally, there is almost no research focusing on the fluctuations of mycotoxins during the co-composting process of silage waste and rapeseed straw under the influence of the added microbial agents. This is an exploration of new materials and new bacterial agents for composting. We hypothesized that inoculating a single bacterial agent would show a limited effect in reducing mycotoxin content and cellulose degradation in composting of moldy silage with rape straw and that these effects could be strengthened by inoculating multiple bacterial agents.

Therefore, the objective of this study was to investigate the effects of three different microbial agents (*Bacillus subtilis*, *Paenibacillus* sp., and *Weissella paramesenteroides*) and their mixtures as inoculants on chemical composition, mycotoxin disappearance, and microbial communities during composting of rape straw and moldy silage. This work may provide new insights into the utilization of silage waste and straw waste and help understand the mechanisms of mycotoxin removal and cellulose degradation by microbial agents in composting processes.

## 2. Materials and Methods

### 2.1. Raw Materials

Both rape straw and moldy silage were obtained from Guanling, Anshun, China (N 105°36′45″, E 25°44′8″). The rape straw had a moisture content of 0.13 ± 0.01, pH of 7.81 ± 0.04, and total nitrogen (TN) content of 10.02 ± 0.72, and the moldy silage had a moisture content of 0.75 ± 0.01, pH of 5.42 ± 0.05, and total nitrogen (TN) content of 18.60 ± 2.31, the main material of which was maize straw. The moldy silage and rape straw were cut to the same size before composting. Otherwise, this study was based on the carbon and nitrogen content of rape straw and silage waste to determine the proportion of mixed compost.

### 2.2. Microbial Agents

*Bacillus subtilis*, provided by Hubei Qiming Bioengineering Co., produces cellulase enzymes in compost to promote cellulose degradation [15]. In addition, *Paenibacillus* sp. came from the School of Resources and Environmental Protection of Guizhou University, and tests have shown that the inoculation of single or multiple cellulose-degrading bacteria can accelerate the decomposition of cellulose in the composting process [16]. Moreover, *Weissella paramesenteroides* was obtained from the College of Animal Science, Guizhou University, and *Weissella paramesenteroides* can promote fermentation and potentially enhance the composting effect. Before composting, *Paenibacillus* sp. was cultured with MRS medium (Aiyan Biology Co., Ltd., Shanghai, China), *Bacillus subtilis* and *Weissella paramesenteroides* were lyophilized bacterial powders, and each inoculant was calculated according to ratios and diluted with distilled water. Each inoculant was sprayed evenly on the compost, mixed by hand, and CK was added in equal volumes of distilled water. Ultimately, the bacterial agents were applied individually or in combination at 1 × 10^7^ cfu/g FW, following the method used in a previous study [17].

### 2.3. Experimental Design

In order to neutralize the acidity of the moldy silage, rape straw was chosen as another raw material for composting, achieving recycling of both agricultural wastes. Based on preliminary tests, mixing rape straw and moldy silage at a ratio of 1:2 with the addition of 500 mL distilled water was ineffective. Therefore, a 1:3 ratio of rape straw to moldy silage, along with 1000 mL of distilled water, was used for composting. The moisture content of the mixed raw material was 0.76 ± 0.01, and the pH was 7.59 ± 0.03.

The experiment was set up with eight treatments, namely CK, B, P, WSP, BP, WSPB, WSPP, and WSPBP. The composting experiment was conducted in composting pots for 60 days with aeration conditions of natural ventilation, turning the piles every week and replenishing water to ensure the microbial activity base. Samples were taken on the 1st, 30th, and 60th days of composting corresponding to warming period, high-temperature period, and ambient period, and 150 g of each sample was divided into two parts; one part was used to test the physicochemical properties, and the other part was stored at −80 °C for genome extraction and identification.

### 2.4. Physicochemical Properties

Temperature variations in the compost were recorded using a multifunction monitor (Multi-Channel Data Loggers, Yokohama, Japan), and temperature variations were recorded once a week, with the final average taken throughout the week. The moisture content of the sample was measured by weighing after thoroughly drying the sample in an oven at 65 °C for 72 h. A total of 20 g of fresh sample was mixed with 180 mL of distilled water and allowed to stand for 24 h. The pH of the filtrate was determined using a pH meter (PHSJ-4F, Shanghai INESA Scientific Instruments Co., Ltd., Shanghai, China). Total nitrogen (TN) was determined using a nitrogen analyzer (Dumatherm DT N PRO, Königswinter, Germany). Ammonia nitrogen (${NH}_{4}^{+}$-N) was measured by colorimetric phenol-sodium hypochlorite. Nitrate nitrogen (${NO}_{3}^{-}$-N) was determined by the standard methods of the ministry of Chinese agriculture [18]. Total carbon (TC) was measured by burning samples in a carbon analyzer at 1100 °C (Multi N/C 3100 and HT 1300, Königswinter, Germany).

### 2.5. Degradation of Hemicellulose, Cellulose and Lignin

The content of hemicellulose, cellulose, and lignin in the composted raw materials were determined by adding detergents to the pulverized samples through fully automatic fiber washing equipment to determine neutral detergent (NDF), acid detergent fiber (ADF), and acid detergent lignin (ADL), and then the lignin content was calculated by incineration through a muffle furnace to obtain the ash content [19]. The mixed raw materials consisted of hemicellulose, cellulose, and lignin, accounting for 19.09 ± 0.07%, 15.52 ± 0.68%, and 35.41 ± 2.23%, respectively.

### 2.6. Analysis of Mycotoxins

Mycotoxins were mainly measured in terms of five indicators, Aflatoxin B1 (AFB1), T-2 toxin, Zearalenone (ZEN), Ochratoxin A (OTA), and Vomitoxin (DON). A 10 g sample was collected and added to 90 mL of deionized water to detect mycotoxins in the compost using an ELISA kit, and finally, the results were calculated using an enzyme marker with a microplate reader of 450 nm. The contents of the five indicators of mycotoxins in the raw mix material were follows: AFB1 was 3.35 ± 0.19 µg /kg and ZEN was 5.18 ± 0.12 µg /kg, OTA was 4.47 ± 0.10 µg /kg, DON was 2.69 ± 0.29 µg /kg, and T-2 toxin was 4.46 ± 0.21 µg /kg.

### 2.7. Microbial Community Structure

The total genomic DNA was extracted from each sample by the CTAB method. PCR amplification and bioinformatics analysis of the samples were performed by Novogene Co., Ltd., Beijing, China. Special primers of 16S (341F and 806R) and ITS (ITS5-1737F and ITS2-2043R) were separately selected to amplify the gene fragments for each silage sample for bacterial and fungal community analysis. High-throughput libraries were sequenced on the PacBio Sequel platform. The raw sequences were processed by Novogene Bioinformatics Co., Ltd. (Beijing, China,), and this was performed on the Magic platform (https://magic.novogene.com/customer/main accessed on 9 September 2024) for RDA maps, alpha diversity, and heatmaps. The raw sequence of the microbial community has been submitted to the NCBI with accession number of PRJNA1126033.

### 2.8. Statistical Analysis

All the tests were performed in triplicates (*n* = 3). One-way analysis of variance (ANOVA) and muti-way analysis of variance were performed using SPSS 26.0 (IBM Corp., Armonk, NY, USA). *p* < 0.05 and *p* < 0.01 were considered statistically significant. Figures were plotted with the Graphpad Prism 9 software and RStudio 2024.12.0 Build 467 software.

## 3. Results

### 3.1. Physical–Chemical Characterization

#### 3.1.1. pH

The pH in all treatment groups followed an increasing and then stabilizing trend. The pH in all treatment groups was lower than in the raw material on day 1, and it increased on days 30 and 60. The lowest pH was 5.54 ± 0.09 for WSPP on day 1, and the highest pH was 7.58 ± 0.14 for the CK group (Figure 1a). On day 30, all the pH values fluctuated between 8.06 ± 0.15 and 8.27 ± 0.14. It remained between 7 and 9 on day 60, so the final pH of the treatment groups was within the recommended standard [20].

#### 3.1.2. Temperature

Fermentation from all treatments occurred at low composting temperatures, similar to the result from Sun et al. [21]. During composting, the ambient temperature ranged from 24.2 to 28.1 °C (Figure 1b). Compared with the ambient temperature, the temperature of all treatment groups reached 30 ± 0.2 °C on day 1, and the highest temperature among them was that of WSPP. Then, their temperatures slightly decreased and increased again and remained at 30 °C on day 29 and then gradually decreased. From day 36, the temperatures of all treatment groups converged with the ambient temperature. However, the temperatures remained high as well as suitable for microbial survival, indicating that the transformation of substances might still have been functioning.

#### 3.1.3. Nitrogen Conversion and C/N

TN in all treatment groups generally showed an increasing trend (Figure 1c), and on day 30, the highest total nitrogen content was in WSPBP (22.93 ± 2.51 g/kg), and the lowest was in P (13.80 ± 0.20 g/kg) and WPS (13.93 ± 0.40 g/kg). The top three TN contents were that of BP (24.47 ± 2.42 g/kg), WSPP (23.77 ± 1.20 g/kg), and WSPBP (23.53 ± 0.87 g/kg) on day 60, which were 65.34%, 60.60%, and 58.99% higher than that of CK, respectively. The increase in TN was associated with the presence of corn straw in the material [22]. It was also possible that nitrogen-fixing bacteria converted nitrogen dioxide or nitrogen gas from the air.

Different from TN, the variation in ${NH}_{4}^{+}$-N is shown in Figure 1d, and it showed a decreasing trend in all treatment groups. On day 30, the highest ${NH}_{4}^{+}$-N content was found in WSPP (10.20 ± 5.51 g/kg) and the lowest was in B (1.47 ± 1.00 g/kg). On day 60, all the treatment groups stabilized, and the lowest ${NH}_{4}^{+}$-N content was in CK (0.3 ± 0.05 g/kg).

Due to the different microbial agents, ${NO}_{3}^{-}$-N had different tendencies in all groups during composting. The contents of ${NO}_{3}^{-}$-N in the treatments with multiple microbial agents were higher than that of the treatments with a single microbial agent (Figure 1e). Throughout composting, ${NO}_{3}^{-}$-N production in WSPBP was the highest, followed by BP, WSPP, and WSPB. On day 60, the ${NO}_{3}^{-}$-N levels in WSPBP (1.21 ± 0.13 g/kg), BP (0.79 ± 0.04 g/kg), and WSPP (0.62 ± 0.12 g/kg) were 97.52%, 96.20%, and 95.16% higher, respectively, than in CK (0.03 ± 0.01 g/kg).

The concentrations of ${NH}_{4}^{+}$-N, C/N tended to decrease in all treatments. However, the treatments with multiple microbial agents had lower C/N levels than those with a single microbial agent. On day 60, C/N levels in WSPP, WSPBP, and BP were 16.55 ± 0.90, 16.23 ± 0.76, and 15.44 ± 1.61. In addition, C/N in CK was 23.10 ± 4.93, indicating multiple microbial agents were beneficial for carbon transformation and nitrogen preservation (Figure 1f).

### 3.2. Analysis of Lignocellulose

Lignocellulose consists of polysaccharides (cellulose, hemicellulose, pectin) with the aromatic polymer lignin. Cellulose, hemicellulose, and lignin represent the primary carbon-containing constituents within biomass, and the interactions between these components can significantly affect the degradation process [23]. Lignin is the most difficult part of agricultural production waste to degrade, so the effective degradation of lignin can achieve ideal results. The changes in the content of cellulose, hemicellulose, and lignin are shown in Figure 2. Overall, for cellulose, hemicellulose, and lignin in the moldy silage and rape straw, their degradation was improved differently by the addition of bacterial agents. As far as hemicellulose is concerned, BP, WSPB, and WSPP had better degradation on day 30, which was 3.72% less than CK, but it was not significant on day 60 for all treatment groups (Figure 2a). As for cellulose, the best degradation was observed for WSPBP, which was 8.57% less than that of CK on day 30, and the same was observed on day 60. However, WSPB and WSPP also ended up with better degradation, at 6.09% and 4.16% less than CK (Figure 2b). In the case of lignin, the most significant reduction was observed in the WSPP group on day 30 (9.57% lower than CK). By day 60, however, the WSPB group surpassed the other treatments with a 7.90% reduction compared to CK, while WSPP retained a moderate reduction of 4.48% (Figure 2c).

The results showed that the cellulose degradation rate of CK after composting was 31.57% and the lignin degradation rate was 19.54%, while the cellulose degradation rate of WSPB was 70.81% and the lignin degradation rate was 41.85%, and the cellulose degradation rate was 58.38% and the lignin degradation rate was 35.02% in WSPP. The addition of bacterial agents improved the degradation of lignin and cellulose more in the co-composting process. WSPB achieved the best lignocellulose degradation in all treatments, followed by WSPP. In addition, different combinations of bacterial agents with different abilities to degrade hemicellulose, cellulose, and lignin could achieve different effects.

### 3.3. Content of Mycotoxins Analysis

Mycotoxin contamination of crops mainly originates from mold infection in the field, and mold management and mycotoxin contamination remain a challenge [24]. During composting, there was variability in mycotoxin levels between treatment groups (Figure 3).

Regarding AFB1, its content was lowest in WSPP on day 30, which was 0.97 µg /kg less than CK. It was also reduced more in BP and WSPBP, which were 0.67 µg /kg less and 0.77 µg /kg less. However, on day 60, WSPP was the only one that was less than CK, which was 0.33 µg /kg less (Figure 3a), indicating that the WSPP could inhibit the production of AFB1. With respect to T-2 toxin, the lowest content of it on day 30 was in WSPB, which was 1.05 µg /kg less than CK, followed by WSPP, which was 0.98 µg /kg less than CK. On the contrary, the lowest amount of T-2 toxin on day 60 was in WSPP, which was 0.32 µg /kg less than CK (Figure 3b), indicating that the WSPP group could inhibit the production of T-2 toxin. In terms of OTA, its lowest content was in WSPP on day 30, which was 0.83 µg /kg less than CK, and all other treatment groups had higher levels than CK. Surprisingly, the lowest level on day 60 was still in WSPP, at 0.45 µg /kg less than CK (Figure 3c). In terms of ZEN, its least content on day 30 was WSPBP, which was 1.22 µg /kg less than CK. The ZEN contents of BP, WSPB, and WSPP were all differently reduced, but on day 60, it was the lowest in WSPP, at 0.68 µg /kg less than CK (Figure 3d). In terms of DON, the lowest content was in WSPBP, at 1.38 µg /kg less than CK on day 30, and the second lowest was in WSPP, at 1.05 µg /kg less than CK. The lowest content was in CK followed by WSPP on day 60. The content in WSPP was 0.20 ug/kg more than CK (Figure 3e). The significance of the five mycotoxins differed between treatments, but they were significantly reduced on different days (*p* < 0.001) (Figure 3f).

The results showed that the addition of microbial agents to the co-compost resulted in a good inhibition of mycotoxin production. However, the one with the best inhibitory effect was WSPP, which reduced the AFB1 content by 64.48%, T-2 toxin content by 65.02%, OTA content by 61.30%, ZEN content by 67.67%, and DON content by 48.33% compared to the original mixed material. It showed the best inhibition of AFB1, T-2 toxin, OTA, and ZEN, while it showed better inhibition of DON.

### 3.4. Analysis of Micro-Ecological Community in Composting

#### 3.4.1. Analysis of Bacterial Community in Composting

The bacterial community changes during composting were analyzed using the results of 16S rDNA high-throughput sequencing. Alpha diversity analysis (Table 1) demonstrated higher bacterial community richness and diversity in all treatment groups. Although CK had higher Shannon, Chao1, and observed species indices than the other groups on day 1, most of the treatment groups were higher than CK in terms of these indices on day 30 and 60.

By classifying the OTUs sequences, 10 dominant phyla (>1.0%) were obtained (Appendix A). As the composting days increased, there were some changes in the dominant flora. On day 30, Proteobacteria and Bacteroidota remained more dominant in all groups, with a large decrease in Firmicutes. The relative abundance of Patescibacteria (2.24 to 8.81%), Fibrobacterota (1.15 to 8.82%), and Bdellovibrionota (5.47 to 7.71%) made it into the top ten. On day 60, there was a small decrease in Proteobacteria in all treatment groups compared to day 30, while there was a tendency for Myxococcota to increase in the treatment group with the addition of the mixed bacterial agent. In particular, Firmicutes had the greatest relative abundance (5.04%) in WSPP. Interestingly, Patescibacteria was the dominant phylum on days 30 and 60.

At the genus level, 20 genera (>1.0%) were dominant (Appendix A). On the day with the highest composting temperature (day 30), the bacterial community radically changed at the genus level, most notably *Cellvibrio* and *Pseudoxanthomonas* were the dominant genera with a large percentage, even reaching 33.91% and 34.46% in WSP and WSPB. Meanwhile, the relative abundance of *TM7a*, *Opitutus*, and *Lacunisphaera* increased. *Cellvibrio*, *Pseudoxanthomonas*, *Paenibacillus*, and *TM7a* levels were 8.15%, 5.28%, 2.22%, and 1.26% in WSPP, respectively. On day 60 compared to day 30, the change in the bacterial community became moderate as the temperature leveled off. There was a trend of a small increase in the relative abundance of *Opitutus* and *Lacunisphaera*, which was consistent with the gate levels.

#### 3.4.2. Analysis of Fungal Community in Composting

The ITS rDNA high-throughput sequencing results were used to analyze the changes in the fungal community of the compost. Alpha diversity analysis (Table 2) showed that the Shannon, Chao1, and observed species indices were lower than those of the other treatment groups in WSPP, whereas the Simpson index was higher, indicating lower fungal abundance and diversity in WSPP.

Classifying the OTUs sequences, 10 dominant phyla were obtained (>1.0%) (Appendix A). On day 30, particularly in BP and WSPB, their primary dominant phylum changed to Aphelidiomycota (53.7% and 62.91%), and the primary dominant phylum changed to Mucoromycota (31.10%) in WSPP. As on day 30 in BP and WSPB, the dominant phylum in all treatment groups on day 60 was Ascomycota (6.26 to 44.27%), Aphelidiomycota (1.44 to 70.97%), Fungi phy Incertae sedis (1.39 to 12.52%), and Basidiomycota (0.46 to 12.00%). In particular, the relative abundance of WSPP at the phylum level was more balanced.

At the genus level, the relative abundance was quite different between days (Appendix A). *Aspergillus*, *Fusarium*, and *Alternaria* are of great interest in terms of fungi that cause disease and produce mycotoxins. On day 1, *Aspergillus* had the highest relative abundance in BP (99.99%), *Fusarium* in WSP (7.00%), and *Alternaria* in CK (3.57%). After composing, the highest relative abundance of *Aspergillus* was WSPB (12.41%), the highest for *Fusarium* was WSPBP (0.54%), and the highest for *Alternaria* was CK (1.25%). Surprisingly, these three fungal genera decreased in WSPP compared to the treatment groups, which was consistent with the phylum.

### 3.5. LEfSe Analysis in Composting

The statistical significance of the differences between samples was analyzed by LEfSe. The evolutionary branch annotations of species with significant differences in composting on different days are shown in Figure 4a,c, and Appendix A. The histogram of the differential species distribution in the compost on day 1 (Appendix A) shows a large difference for all groups. On day 30 (Figure 4b), BP contained four high-abundance branches, Bacilli (Class) and Microscillaceae (Family), and WSPBP contained four rich branches, including Actinobacteriota (Phylum) and Actinobacteria (Class), whereas CK only contained one, Xanthomonadales (Order). On day 60 (Figure 4d), the highly abundant species were Bacilli (Class) and Paenibacillales (Order) (three species in total) in WSPP, Myxococcota (Phylum) and Polyangia (Class) (five species in total) in WSPBP, Verrucomicrobiota (Phylum) and Verrucomicrobiae (Class) (five species in total) in P, and Fibrobacterota (Phylum) and Fibrobacteria (Class) (six species in total) in CK.

### 3.6. Correlation Analysis with Environmental Factors

Spearman’s correlation analysis with the Mantel test revealed the association of environmental factors, insoluble fiber, and mycotoxins with the bacterial community and fungal community in the compost (Figure 5). On day 30, both the bacterial community and fungal community had a significantly positive correlation regarding TN and ${NO}_{3}^{-}$-N (*p* < 0.05). Notably, the fungal community showed a positive correlation with cellulose and hemicellulose (*p* < 0.01) as well as with mycotoxins. In contrast, the fungal community was negatively correlated with most of the mycotoxins (Figure 5a). On day 60, the bacterial and fungal communities remained significantly positively correlated with cellulose and TN (*p* < 0.05) (Figure 5b). *Paenibacillus*, *Luteimonas*, *Pseudoxanthomonas*, *Pseudomonas*, and *Bdellovibrio* were positively correlated with environmental factors, especially TN, ${NH}_{4}^{+}$-N, cellulose, and ZEN, which were very significant at day 30 (*p* < 0.05) (Figure 5c). *Paenibacillus*, *Pseudoxanthomonas*, *BIrii41*, and *TM7a* were highly correlated with environmental factors, especially TN, ${NH}_{4}^{+}$-N, cellulose, and lignin, which were very significant at day 60 (*p* < 0.05) (Figure 5d).

Redundancy analysis (RDA) and Spearman’s correlation analysis can visualize the correlation between microbial communities and environmental factors as well as better understand the changes in compost properties with different treatment groups. The results of the RDA showed that the correlation between environmental factors and bacterial and fungal communities varied greatly among the different treatments (Figure 6 and Appendix A). Interestingly, on day 30, WSPP behaved quite differently from the other treatment groups in that its bacterial community was positively correlated with TN and ${NH}_{4}^{+}$-N, whereas it was negatively correlated with mycotoxins, hemicellulose, and cellulose as well as its fungal community (Figure 6a,b and Appendix A). Different from day 30, the bacterial community in WSPP was negatively correlated with a large proportion of mycotoxins on day 60. However, the fungal community in WSPP was negatively correlated with both insoluble fiber and mycotoxins (Figure 6c,d and Appendix A). The results showed that the microbial communities inoculated with different bacterial agents produced different effects on environmental factors. Notably, inoculation with *Weissella paramesenteroides* and *Paenibacillus* sp. in the microbial communities contributed to an increased nitrogen content and reduced mycotoxins. Bacteria of the genus *Paenibacillus* sp. have also been identified as strains with the potential to biodegrade lignocellulose through the secretion of enzymes such as amylases, cellulases, hemicellulases, lipases, and other ligninolytic enzymes, and they show high humification indices, and high activities of cellulase, beta-glucosidase, and alkaline phospho-monoesterase were also achieved [25].

Correlation heat map analysis further revealed the association between environmental factors, lignocellulose, and mycotoxins with the microbial community in the compost. The effect of the microbial community on insoluble fiber and mycotoxins was different on different days and with different environmental factors (Figure 7). The correlation between the microbial communities at the genus level and environmental factors changed over time (Figure 7c–f). Overall, only *Paenibacillus* showed a significant correlation (*p* < 0.05) with hemicellulose and cellulose in the bacterial community, and *IS-44*, *Sumerlaea*, *Moheibacter*, and *unidentified BIrii44* were also correlated with mycotoxins and cellulose, indicating that their activities could contribute to cellulose and hemicellulose degradation as well as mycotoxin reduction (Figure 7a). In addition, among the fungal community, the fungus with a significant positive correlation with the five mycotoxins was mainly *Fusarium* (Figure 7b).

### 3.7. Relationship Among Bacterial Community, Paenibacillus sp., TN, Mycotoxins, and Lignocellulose

Among the inoculations of bacterial agents in the compost, only *Paenibacillus* sp. was dominant in the bacterial community, and it was thus important to study. Based on the PLSPM model, this study evaluated the complex influence of the bacterial community and *Paenibacillus* sp. on the increase in TN, the reduction in mycotoxins, and lignocellulose degradation in the compost, as shown in Figure 8. The goodness-of-fit (GOF) index was 0.66 in this model, which means the model had good prediction performance [26]. The results indicated that the addition of *Paenibacillus* sp. significantly enhanced the increase in TN (R = 0.48, *p* < 0.01) and lignocellulose degradation (R = 0.08, *p* < 0.001), and it presented a significant influence on the reduction in mycotoxins (R = 0.35, *p* < 0.01). The bacterial community only enhanced the increase in TN, yet it presented as non-significant (R = 0.24, *p* > 0.05). This result confirmed that the addition of *Paenibacillus* sp. was one of the main pathways for the removal of mycotoxins and lignocellulose degradation in this study, but there was a competitive relationship between *Paenibacillus* sp. and the bacterial community in the compost (R = −0.27, *p* > 0.05). Overall, the PLSPM model revealed that *Paenibacillus* sp. played an important role in regulating the complicated relationships among TN, mycotoxins, and lignocellulose, and it could improve the total nitrogen, reduce mycotoxins, and degrade lignocellulose.

## 4. Discussion

The aim of this research was to explain the effect of different exogenous bacterial agents and their combinations on the transformation of mycotoxins and the degradation of lignocellulose, which are not common in the co-composting of silage waste with rapeseed straw. In terms of composting characteristics, lignocellulose degradation, mycotoxin abundance, and microbial community changes, the addition of *Weissella paramesenteroides* and *Paenibacillus* not only accelerated the degradation of hemicellulose and cellulose but also inhibited mycotoxin production, resulting in a compost product that presents less risk to animal ingestion and a lower environmental risk.

The pH influenced the microbial activity and composting process, and the degradation of nitrogenous substances and the generation of ammonium salt resulted in a gradual increase in pH during composting [27]. Some studies have shown that microorganisms exhibit the highest degradation activity in the pH range of from 7 to 9 [28]. In addition, temperature is one of the most important indicators in composting and reflects the microbial activity and decomposition rate [29]. During composting, the ambient temperature affected the basal metabolism of the microorganisms, and the microbial metabolism caused an increase in the temperature. ${NH}_{4}^{+}$-N decreased rapidly in all groups due to volatilization and nitrification, but TN and ${NO}_{3}^{-}$-N increased gradually, which was due to the greater level of nitrification than volatilization and low-temperature composting reducing ammonia volatilization [30]. ${NH}_{4}^{+}$-N in WSPP slowly decreased in 30 days, while TN slowly increased inversely, which may indicate that the addition of *Weissella paramesenteroides* and *Paenibacillus* enhanced the level of nitrogen fixation. The results were consistent with the findings of a previous study, in which *Paenibacillus* strengthened its fixation of N [31]. TN in BP, WSPP, and WSPBP after composting was higher than in the other groups, indicating that *Paenibacillus* had a strong nitrogen retention effect even under the influence of mycotoxins.

Microorganisms, as drivers of material conversion, were highly relevant to the degradation of lignocellulose. The combined action of *Paenibacillus*, *Pseudoxanthomonas*, *Cellvibrio*, *Bacillus*, *BIrii41*, *TM7a*, *Luteimonas*, and other bacteria in the bacterial community could efficiently degrade hemicellulose, cellulose, and lignin [32]. In addition, other studies have shown that some fungi also demonstrated outstanding degradation effects on hemicellulose [33]. Lignocellulose was well degraded in WSPB and WSPP, which may have been due to the enhance lignocellulose degradation by *Bacillus* and *Paenibacillus*, in agreement with the findings of Edgar et al. [25]. The fungal community richness and diversity were lower in WSPP and WSPBP, which may be attributed to *Weissella paramesenteroides* contributing to fermentation and inhibiting the growth of harmful microorganisms [34]. However, Qi et al. reported that competition within the microbial community was severely affected by mycotoxins [35]. Excess mycotoxins reduced the diversity, abundance, population size, and metabolic activity of bacterial communities [36]. The rapid reduction in Ascomycota in WSPB and WSPP during the warming period also led directly to a reduction in the pathogenic bacteria *Aspergillus* and *Fusarium*. It is possible that bacterial agents enhanced the competitiveness of the bacterial community against the fungal community. Some studies have shown that *Bacillus* had a strong biodegradation activity against *Fusarium* mycotoxins [37], and *Paenibacillus* shared the same properties [38].

Based on LEfse analysis, we found that all groups had significant differences in the dominant microorganisms, which may be closely related to the type of bacterial agents used. Unlike animal manure composting, the co-composting of the two waste agricultural by-products produced fewer pathogens than manure [39]. As the temperatures rise in the active phase, high-temperature resistant pathogens thrive, while some pathogens experience thermal stress, resulting in their lower abundance [40]. *Bacillus subtilis* produces mycobacteriostats [41], and *Paenibacillus* produces antibiotics [42]. From day 30 to day 60, only WSPP had *Paenibacillus*. Integrating these findings with our observations, we propose a potential mechanism for composting with the addition of different bacterial agents. Inoculation with bacterial agents not only improves the competitiveness of the bacterial community, but its metabolites may also participate in potential metabolic regulation, potentially revealing novel mechanisms of lignocellulose and mycotoxin reduction.

Microbial interactions have implications for natural environmental modification and biodiversity conservation [43]. The pH and moisture content in the composting process were important factors that influenced the community structure, and some of the bacteria in the community also played a role in regulating the community structure. *Patescibacteria*, known as microbial dark matter, is widespread in nature and also in composting, where it survives by parasitizing other host bacteria and has the ability to regulate the structure of the microbial community [44]. Microbial interactions accelerated the fixation of the microbial community structure. Spearman’s correlation analysis further validated that microbial interactions could enhance the lignocellulose degradation. *Pseudoxanthomonas* has been reported to combine with other bacteria to degrade cellulose and lignin to provide a carbon source for survival [45]. Inoculation with *Weissella paramesenteroides* and *Paenibacillus* sp. accelerated the microbial metabolism, material decomposition, and mycotoxin reduction. *Paenibacillus* sp. could promote crop growth directly via biological nitrogen fixation, phosphate solubilization, and the production of the phytohormone indole-3-acetic acid (IAA), and it could also offer protection against insect herbivores and phytopathogens, including bacteria, fungi, nematodes, and viruses [46]. They or their isolated antimicrobial compounds could therefore be useful in controlling phytopathogenic microorganisms, leading to a lower usage of chemical biocides, which can have negative environmental effects [47]. Based on the PLSPM model, *Paenibacillus* sp. was largely involved in nitrogen fixation [48], lignocellulose decomposition [49], and the secretion of antifungal compounds [50].

In this study, the addition of the different bacterial agents resulted in different levels of degradation of lignocellulose and different levels of inhibition of mycotoxins. *Weissella paramesenteroides*, as a functional bacterium, is critical for maintaining a balanced microbial community [51]. Lignocellulose-degrading bacteria (*Flavobacterium*, *Pseudoxanthomonas*, *Cellvibrio*, *Bacillus*, *Paenibacillus*, etc.) gradually increased and became dominant, which could improve the composting efficiency, and *Paenibacillus* could effectively inhibit mycotoxin production and reduce environmental and health risks [52]. These findings could provide a theoretical basis for the degradation of lignocellulose and mycotoxins in moldy silage and rape straw compost. In addition, the results of this composting experiment in other climates or with other bacterial agents require more attention, and this study focused on the harmless utilization of moldy silage.

## 5. Conclusions

This study revealed that the inoculation of *Weissella paramesenteroides* and *Bacillus subtilis* could effectively enhance the degradation rate of cellulose by 39.24% and lignin by 22.31%, and the inoculation of *Weissella paramesenteroides* and *Paenibacillus* sp. could enhance the degradation rate of cellulose by 26.75% and lignin by 15.48% and reduce the levels of mycotoxins (*p* < 0.05; e.g., AFB1 reduced by 64.48%, T-2 toxin reduced by 65.02%, OTA reduced by 61.30%, ZEN reduced by 67.67%, and DON reduced by 48.33%). Inoculation with *Paenibacillus* sp. and other bacteria increased the total nitrogen content by 48.34% to 65.52% through enhancing the microbiological activity. Inoculation with WSPP enhanced the composting efficiency while composting at low temperatures, which led to the enhancement of organic decomposition and nitrogen fixation. More importantly, WSPP combined with other bacteria closely strengthened the mycotoxin reduction and lignocellulose degradation. Thus, the bacterial agents show potential for the aerobic co-composting of moldy silage with rape straw, providing valuable reference information for future composting management and public health protection and a better direction for the recycling of agricultural waste.

## Figures and Tables

**Figure 1 microorganisms-13-00677-f001:**
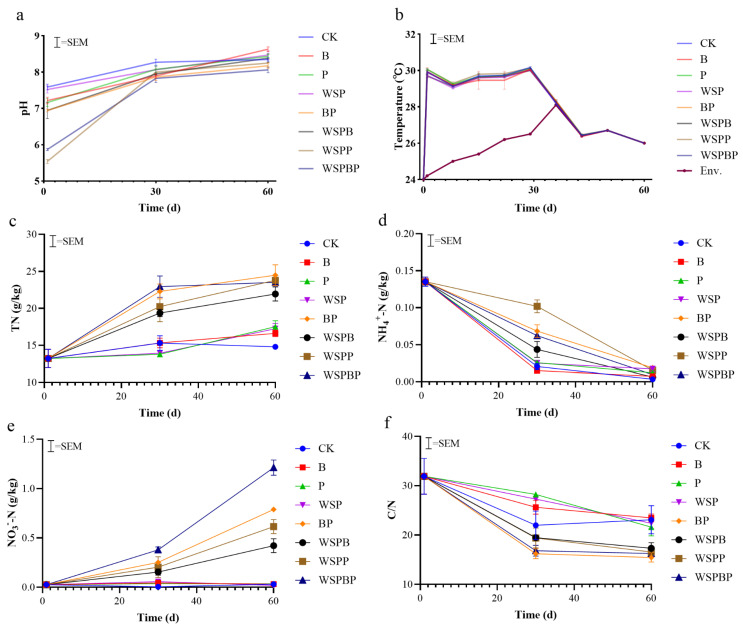
Effect of different microbial agents on (**a**) pH, (**b**) temperature (**c**) TN, total nitrogen, (**d**) ${NH}_{4}^{+}$-N, ammoniacal nitrogen, (**e**) ${NO}_{3}^{-}$-N, nitrate nitrogen, and (**f**) C/N, carbon/nitrogen ratio during composting. CK, without bacterial agents in the co-compost; B, inoculation with *Bacillus subtilis* in the co-compost; P, inoculation with *Paenibacillus* sp. in the co-compost; WSP, inoculation with *Weissella paramesenteroides* in the co-compost; BP, inoculation with *Bacillus subtilis* and *Paenibacillus* sp. in the co-compost; WSPB, inoculation with *Weissella paramesenteroides* and *Bacillus subtilis* in the co-compost; WSPP, inoculation with *Weissella paramesenteroides* and *Paenibacillus* sp. in the co-compost; WSPBP, inoculation with *Weissella paramesenteroides*, *Bacillus subtilis* and *Paenibacillus* sp. in the co-compost. Abbreviations in the following figures mean the same.

**Figure 2 microorganisms-13-00677-f002:**
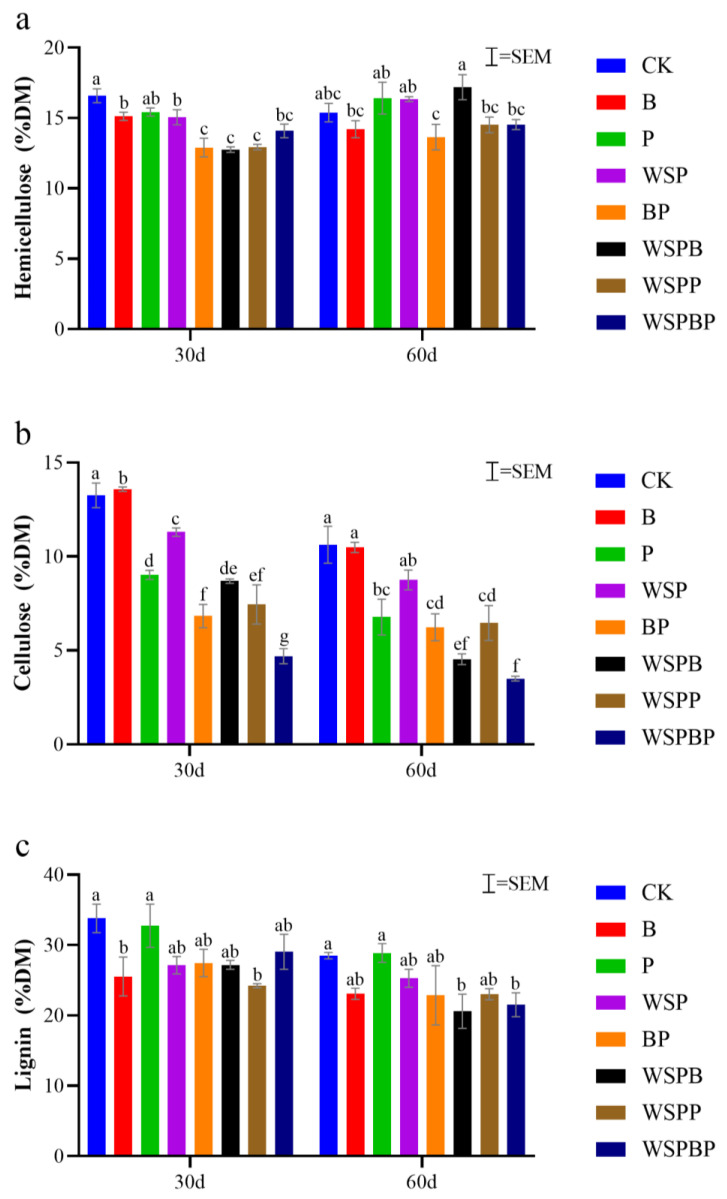
Effect of different microbial agents on the change in lignocellulose, including (**a**) hemicellulose, (**b**) cellulose, and (**c**) lignin, during composting. Different letters indicate significant differences among treatments on the same day (*p* < 0.05). Equal letters indicate insignificant differences (*p* > 0.05).

**Figure 3 microorganisms-13-00677-f003:**
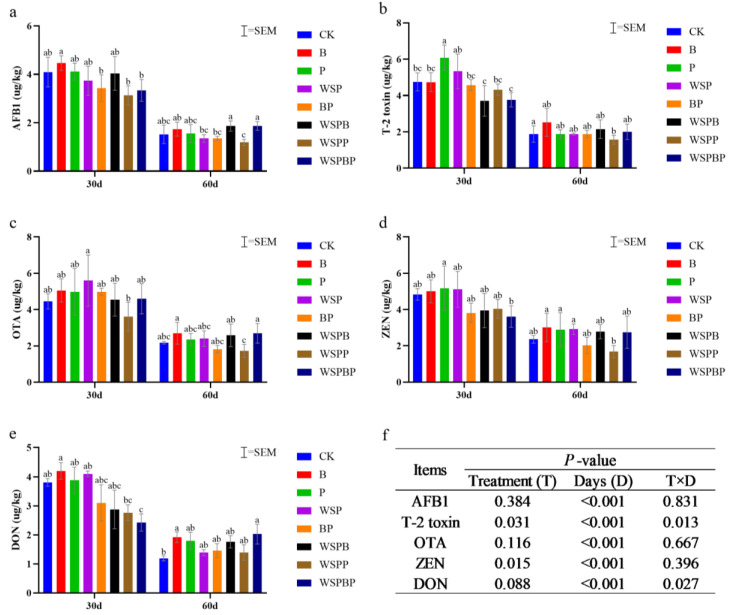
Content results of mycotoxin changes in different treatments during compost. (**a**) Aflatoxin B1 (AFB1), (**b**) T-2 toxin, (**c**) Ochratoxin A (OTA), (**d**) Zearalenone (ZEN), and (**e**) Vomitoxin (DON). (**f**) Significant differences of mycotoxin interaction effects according to the LSD test. Different letters indicate significant differences among treatments on the same day (*p* < 0.05). Equal letters indicate insignificant differences (*p* > 0.05).

**Figure 4 microorganisms-13-00677-f004:**
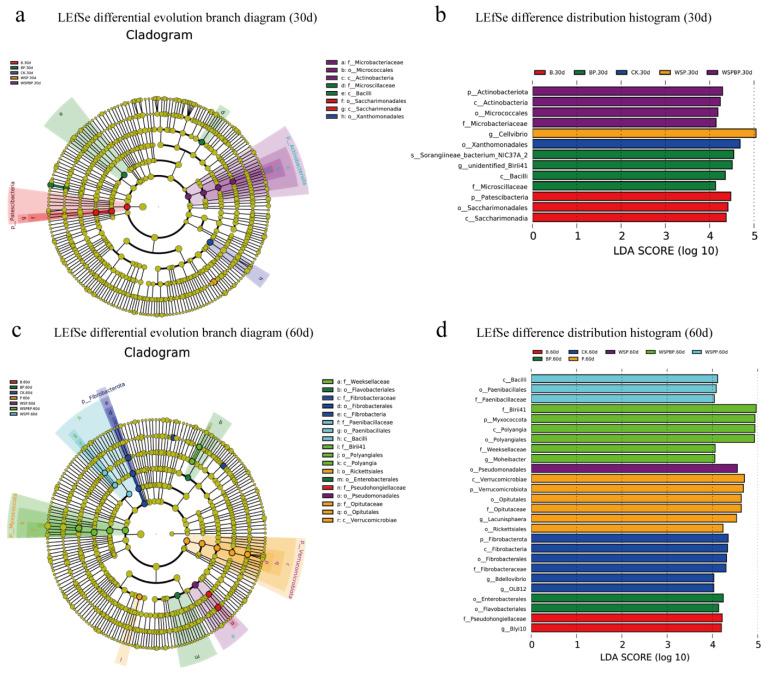
LEfSe analyses of the species with significant differences between groups in composting on day 30 and day 60, including (**a**) LEfSe differential evolution branch diagram on day 30, (**b**) LEfSe difference distribution histogram on day 30, (**c**) LEfSe differential evolution branch diagram on day 60, and (**d**) LEfSe difference distribution histogram on day 60. Note: In the evolutionary branch diagram, the starting level is Phylum and the ending level is Species. Yellow nodes indicate no significant difference between the groups.

**Figure 5 microorganisms-13-00677-f005:**
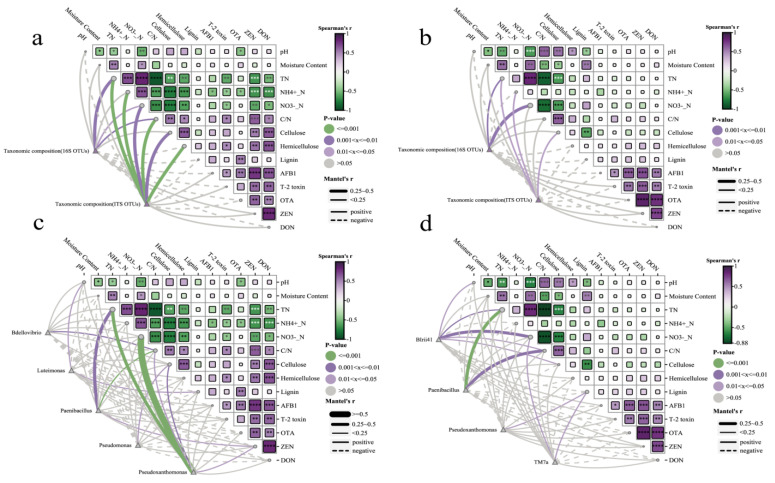
Pairwise comparisons of environmental factors are shown, with the color gradient indicating Spearman’s correlation coefficients at (**a**) day 30 of the taxonomic community (16S and ITS), (**b**) day 60 of the taxonomic community (16S and ITS), (**c**) day 30 of the significant bacteria and inoculated bacteria (genus level), and (**d**) day 60 of the significant bacteria and inoculated bacteria (genus level). Taxonomic community (16S and ITS) composition and significant bacteria were related to each environmental factor by partial Mantel tests. Edge width corresponds to Mantel’s r statistic for the corresponding distance correlations. * Significant at *p* < 0.05; ** *p* < 0.01; *** *p* < 0.001; **** *p* < 0.001.

**Figure 6 microorganisms-13-00677-f006:**
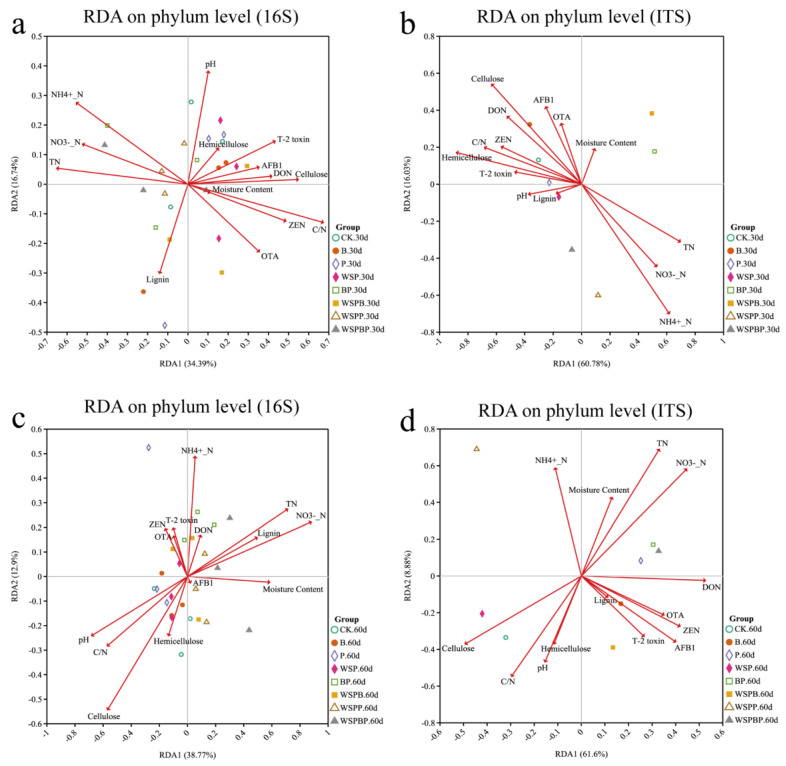
Redundancy analysis (RDA) of the correlation among the microbial communities and physicochemical parameters based on bacterial community (16S) and fungal community (ITS) at the phylum level, including (**a**) RDA on phylum level (16S) on day 30, (**b**) RDA on phylum level (ITS) on day 30, (**c**) RDA on phylum level (16S) on day 60, and (**d**) RDA on phylum level (ITS) on day 60.

**Figure 7 microorganisms-13-00677-f007:**
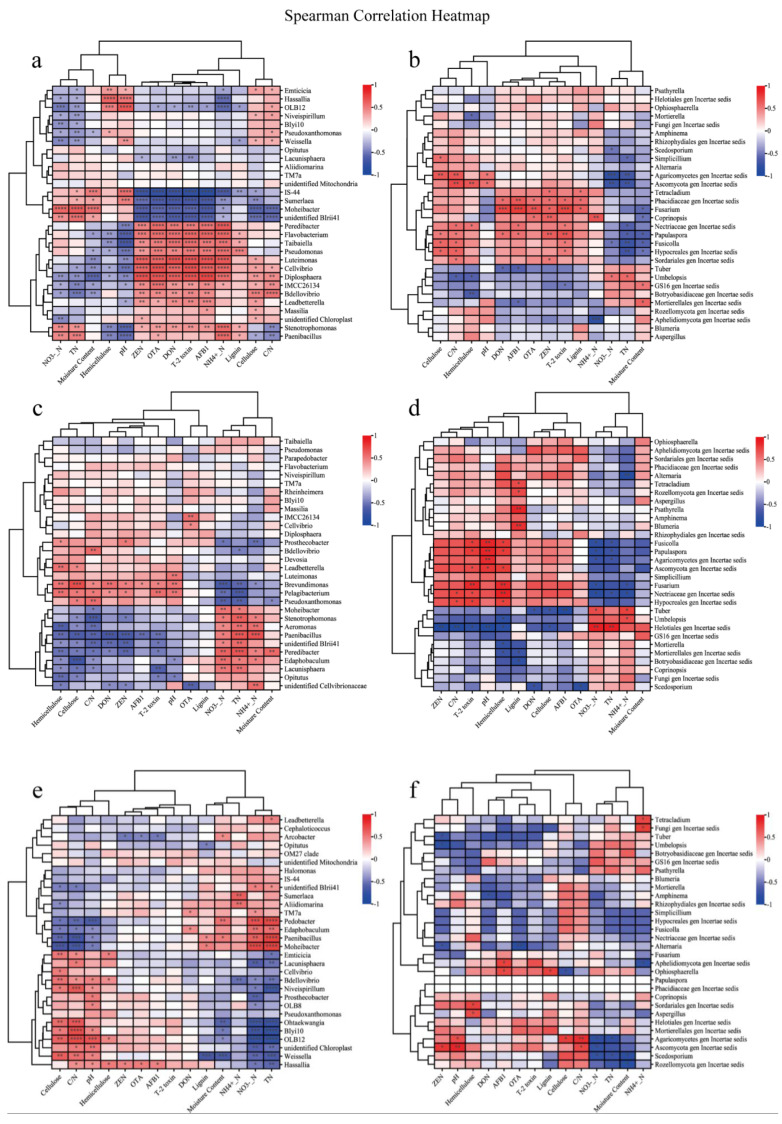
Heatmap of Spearman’s correlation between physicochemical properties, lignocellulose, mycotoxins, and primary microbial genera during the composting process. * Significant at *p* < 0.05; ** *p* < 0.01; *** *p* < 0.001; **** *p* < 0.001. (**a**) Bacterial community, (**b**) fungal community, (**c**) bacterial community on day 30, (**d**) fungal community on day 30, (**e**) bacterial community on day 60, and (**f**) fungal community on day 60.

**Figure 8 microorganisms-13-00677-f008:**
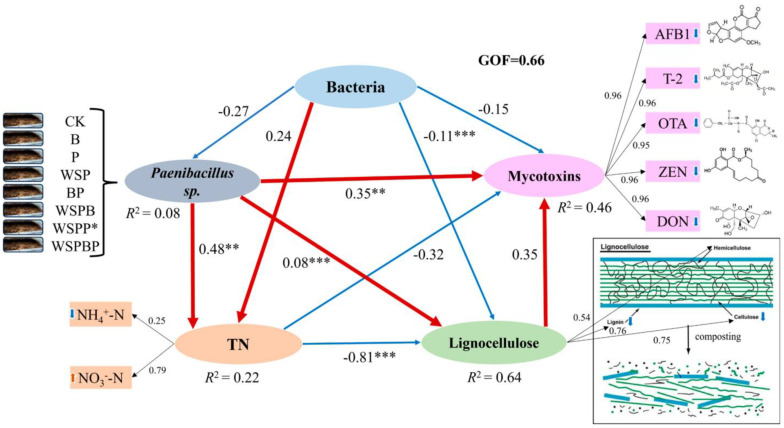
Analysis results of PLSPM in composting. The red and blue lines present the positive and negative direct effects, respectively. Numbers adjacent to arrows are path coefficients. The proportions of the variance explained for each variable are denoted by R^2^ values. Significance levels are indicated as follows: * *p* < 0.05, ** *p* < 0.01, *** *p* < 0.001.

**Table 1 microorganisms-13-00677-t001:** Alpha diversity indexes of bacteria community in moldy silage compositing with rape straw, inoculated without (CK) or with *Bacillus subtilis* (B), *Paenibacillus* sp. (P), *Weissella paramesenteroides* (WSP), and their combinations (B + P, BP; WSP + B, WSPB; WSP + P, WSPP; and P + B + WSP, WSPBP), at day 1, 30, and 60. Different lowercase letters indicate significant differences between treatments (*p* < 0.05). Different capital letters indicate significant differences between treatments (*p* < 0.05). Equal letters indicate insignificant differences (*p* > 0.05).

Index	Composting Period	Treatment (T)								SEM	*p*-Value		
		CK	B	P	WSP	BP	WSPB	WSPP	WSPBP		T	D	T × D
OTUs	Day 1	638 Ca	554 Bab	553 Cab	596 Ca	533 Cab	449 Cb	531 Cab	563 Bab	31.889	0.001	<0.001	0.170
	Day 30	906 Bc	1155 Aab	909 Bc	920 Bc	943 Bbc	732 Bc	953 Bbc	1198 Aa				
	Day 60	1017 A	1254 A	1076 A	1051 A	1165 A	1012 A	1034 A	1166 A				
Chao1	Day 1	766.36 Ca	673.20 Bab	697.61 Cab	705.09 Cab	660.90 Cabc	505.19 Cc	577.38 Cbc	674.26 Bab	40.147	0.001	<0.001	0.295
	Day 30	1087.01 Bbc	1503.41 Aa	1024.59 Bc	1115.23 Bbc	1138.34 Bbc	877.27 Bc	1112.50 Bbc	1401.07 Aab				
	Day 60	1226.58 A	1590.83 A	1304.99 A	1292.61 A	1494.61 A	1187.32 A	1205.31 A	1423.85 A				
Coverage	Day 1	0.998	0.998	0.998	0.998	0.998	0.999	0.999	0.998	0.0001	0.002	<0.001	0.786
	Day 30	0.997	0.996	0.998	0.997	0.998	0.998	0.998	0.997				
	Day 60	0.997	0.996	0.997	0.997	0.996	0.997	0.997	0.996				
Shannon	Day 1	6.33 Ba	5.56 Bab	5.45 Bab	5.63 Cab	5.98 Cab	5.08 Cb	5.52 Bab	5.96 Cab	0.112	0.001	<0.001	0.005
	Day 30	7.30 Ab	7.72 Aab	7.07 Abc	6.54 Bcd	7.07 Bbc	6.23 Bd	7.73 Aab	8.14 Aa				
	Day 60	7.22 Aab	7.90 Aa	7.12 Ab	7.48 Aab	7.60 Aab	7.29 Aab	7.23 Aab	7.04 Bb				

**Table 2 microorganisms-13-00677-t002:** Alpha diversity indexes of fungal community in moldy silage compositing with rape straw, inoculated without (CK) or with *Bacillus subtilis* (B), *Paenibacillus* sp. (P), *Weissella paramesenteroides* (WSP), and their combinations (B + P, BP; WSP + B, WSPB; WSP + P, WSPP; and P + B + WSP, WSPBP), at day 1, 30, and 60. Different lowercase letters indicate significant differences between treatments (*p* < 0.05). Different capital letters indicate significant differences between treatments (*p* < 0.05). Equal letters indicate insignificant differences (*p* > 0.05).

Index	Composting Period	Treatment (T)								SEM	*p*-Value		
		CK	B	P	WSP	BP	WSPB	WSPP	WSPBP		T	D	T × D
OTUs	Day 1	348 Ca	293 Bb	87 Cf	101 Cef	10 Cg	168 Ac	112 Be	129 Bd	23.906	0.001	<0.001	0.124
	Day 30	477 Bd	365 Ae	532 Ac	824 Aa	131 Bf	167 Af	144 Af	618 Ab				
	Day 60	512 Aa	118 Cd	119 Bd	385 Bb	159 Ac	141 Bcd	118 Bd	137 Bcd				
Chao1	Day 1	367.83 Ca	314.14 Bb	92.50 Cf	102.43 Cef	10.00 Cg	171.21 Bc	112.00 Be	131.33 Bd	26.686	0.001	<0.001	0.237
	Day 30	488.54 Bd	419.81 Ae	553.50 Ac	951.10 Aa	148.27 Bh	190.00 Af	153.23 Ag	665.19 Ab				
	Day 60	568.95 Aa	123.14 Cf	130.25 Be	419.17 Bb	167.57 Ac	144.88 Cd	120.33 Bg	145.67 Bd				
Coverage	Day 1	0.999	0.999	1.000	1.000	1.000	1.000	1.000	1.000	0.0001	0.001	<0.001	0.115
	Day 30	0.999	0.998	0.999	0.997	0.999	0.999	1.000	0.998				
	Day 60	0.998	1.000	1.000	0.999	1.000	1.000	1.000	1.000				
Shannon	Day 1	5.77 Bb	6.15 Aa	3.48 Bg	4.61 Cd	0.88 Ch	3.77 Af	5.20 Ac	4.50 Be	0.190	0.001	<0.001	0.005
	Day 30	6.37 Aa	3.24 BCc	6.52 Aa	7.03 Aa	3.10 Ac	3.20 Bc	4.96 Ab	6.70 Aa				
	Day 60	5.94 Ba	3.73 Bb	2.90 Cc	5.78 Ba	2.82 Bc	3.06 Bc	4.12 Bb	2.72 Cc				

## Data Availability

The original contributions presented in the study are included in the article; further inquiries can be directed to the corresponding author.

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
