# Peer review of "Enhancing Lignocellulose Degradation and Mycotoxin Reduction in Co-Composting with Bacterial Inoculation"

_microorganisms, 2025, doi:10.3390/microorganisms13030677_

Round 1
Reviewer 1 Report
Comments and Suggestions for Authors
Dear Editor, Dear Authors
I was invited to evaluate the manuscript « The inoculation of bacterial agents enhances lignocellulose degradation and mycotoxins reduction during aerobic co-composting of mycotoxins-contaminated silage and rape straw »
In this study, the authors investigated the effect of bacteria inoculation enhancing cellulose and mycotoxins degradation in silage and rape straw. For that, the authors exposed mycotoxins-contaminated silage and rape straw to different bacterial agents and their combinations. Provided data show that the inoculation of Weissella paramesenteroides and Bacillus subtilis causes degradation of cellulose by 39.24% and lignin by 22.31%. In comparison, the inoculation of W. paramesenteroides and Paenibacillus sp. degrades cellulose by 26.75% and lignin by 15.48%, and reduces the levels of mycotoxins (AFB1 down by 64.48%, T-2 toxin 65.02%, OTA 61.30%, ZEN 67.67%, DON 48.33%). In parallel, inoculation was found to increase total nitrogen by 48.34-65.52% through enhancing microbiological activity. Conclusions of the authors are that Paenibacillus sp. combined with other bacteria could increase compost efficiency and reduce mycotoxins presence for better and safe utilization of waste agricultural by-products.
I found the study interesting and very well designed/conducted
Please find some comments :
- Fig 1 : please give more information in the figure legend, including the meaning of each term.
- Same for fig 2 and 3 legends : more details please, including the meaning of letters used in the figure
- Fig 4 and 7 : may be good to make it bigger to make it easier to read
- Fig 3 shows the reduction in mycotoxins. Do the authors looked at the identification of the degradation product(s) ? Reduction can be either binding and/or degradation. But if degradation occurs, at least some metabolites should be found. Can the authors try to measure their production and identify them ?
regards
Author Response
Dear reviewer 1.
Thank you very much for taking the time to review this manuscript. Please find the detailed responses below and the corresponding revisions/corrections highlighted/in track changes in the re-submitted files.
Comments 1: Fig 1: please give more information in the figure legend, including the meaning of each term. Same for Figure 2 and 3 legends: more details please, including the meaning of letters used in the figure.
Response 1: Thank you for pointing this out. We agree with this comment. Therefore, we have made the following changes in red, please see page 6 of the revised manuscript, lines 216–223, and page 7, lines 252–253, and page 9, lines 287-286.
Comments 2: Fig 4 and 7: may be good to make it bigger to make it easier to read.
Response 2: Agree. We've done our best to make this chart larger. The pixels of these image are high enough to support readers to download and view them enlarged on their own.
Comments 3: Fig 3 shows the reduction in mycotoxins. Do the authors looked at the identification of the degradation product(s)? Reduction can be either binding and/or degradation. But if degradation occurs, at least some metabolites should be found. Can the authors try to measure their production and identify them?
Response 3: Thank you for pointing this out. We sincerely appreciate your insightful question regarding the identification of degradation products. You rightly point out that mycotoxin reduction could result from either binding or degradation processes. In the current study, we focused primarily on quantifying the efficiency of mycotoxin reduction efficiency and analyzing key physicochemical parameters during co-composting. Unfortunately, metabolomic analysis of degradation products was not performed due to limitations in analytical instrumentation and research scope. In future work, we plan to collaborate with laboratories equipped for metabolite profiling to identify specific degradation pathways. We thank you for highlighting this critical aspect and welcome suggestions on methods to characterize metabolites in composting systems.

Reviewer 2 Report
Comments and Suggestions for Authors
The study investigates the effect of bacterial inoculation on the degradation of lignocellulose and reduction of mycotoxins during the aerobic co-composting of mycotoxin-contaminated silage and rape straw. The authors analyze different bacterial agents, including Weissella paramesenteroides, Bacillus subtilis, and Paenibacillus sp., and their combinations. The results suggest bacterial inoculation enhances lignocellulose degradation, increases nitrogen retention, and reduces mycotoxin levels, mainly when using Weissella paramesenteroides and Paenibacillus sp.. The study provides insights into the potential of microbial agents for improving composting efficiency and reducing agricultural waste toxicity.
Please find below my comments regarding each section of the paper.
Title: The title is too long and complex. Consider simplifying it while maintaining clarity (e.g., “Enhancing Lignocellulose Degradation and Mycotoxin Reduction in Co-Composting with Bacterial Inoculation”).
Abstract: Quantitative results are dense; use structured sentences for clarity. The practical significance of findings (e.g., impact on waste management) is not well emphasized.
Introduction: The study’s novelty is not clearly stated—how does it improve upon existing composting research? The claim that "50% of silage is contaminated with mycotoxins" lacks a proper citation. The research hypothesis is buried in the last paragraph—move it earlier for clarity.
Materials and methods: Replication details are missing—how many replicates were used for microbial sequencing? The bacterial inoculation process needs more details—how were the agents prepared and applied? The description of the control group (CK) is vague—does it include only uninoculated compost? Statistical methods (ANOVA) are not fully justified—no mention of post-hoc tests or assumption checks.
Results: Figures and tables need improvement for clarity:
- Figure 1 lacks clear differentiation between treatments. Plus the legend is not explained in the figure title.
- Statistical validation is incomplete:
- Confidence intervals (CIs) and effect sizes are missing.
- No mention of assumption checks for ANOVA (e.g., normality, homogeneity of variance).
Discussion
- Some conclusions are speculative and lack direct evidence:
- The claim that Paenibacillus contributes to nitrogen fixation needs supporting enzymatic or genomic data.
- The role of temperature in microbial survival should be linked to composting efficiency.
- The study does not compare results with previous bacterial inoculation studies—how do findings align with existing research?
- Scalability for large-scale composting is not discussed—what are the practical limitations?
Figures and tables: Legends should specify statistical significance. Overlapping error bars make interpretation difficult. Supplementary tables should include raw mycotoxin degradation data over time.
Language and citation quality
- Some sentences are too long and complex—restructure for clarity.
- Reference formatting is inconsistent (e.g., placeholder text in citations).
- Some statements lack citations—ensure all claims are properly referenced.
Comments on the Quality of English Language
The English could be improved to more clearly express the research.
Author Response
Dear reviewer 2.
Thank you very much for taking the time to review this manuscript. Please find the detailed responses below and the corresponding revisions/corrections highlighted/in track changes in the re-submitted files.
Comments 1: Title: The title is too long and complex. Consider simplifying it while maintaining clarity (e.g., “Enhancing Lignocellulose Degradation and Mycotoxin Reduction in Co-Composting with Bacterial Inoculation”).
Response 1: We agree with this comment. Therefore, We've changed the title of the manuscript to “Enhancing Lignocellulose Degradation and Mycotoxin Reduction in Co-Composting with Bacterial Inoculation”.
Comments 2: Abstract: Quantitative results are dense; use structured sentences for clarity. The practical significance of findings (e.g., impact on waste management) is not well emphasized.
Response 2: Thank you for pointing these out. We have revised the quantitative results to be in structured sentences and emphasized the practical significance of findings for waste management. Please see page 1 of the revised manuscript, lines 19–23, and lines 25–29, which are showen in red.
Comments 3: Introduction: The study’s novelty is not clearly stated—how does it improve upon existing composting research? The claim that "50% of silage is contaminated with mycotoxins" lacks a proper citation. The research hypothesis is buried in the last paragraph—move it earlier for clarity.
Response 3: Thank you for pointing these out. Therefore, we have added a description of the novelty of the study. The claim that "50% of silage is contaminated with mycotoxins" has been revised to “A survey on the worldwide occurrence of mycotoxins revealed that 81% of 7,049 live-stock feed samples collected from the Americas, Europe, and Asia were positive for at least one mycotoxin.” The research hypothesis have moved to lines 79-81. All changes are shown in red.
Comments 4: Materials and methods: Replication details are missing—how many replicates were used for microbial sequencing? The bacterial inoculation process needs more details—how were the agents prepared and applied? The description of the control group (CK) is vague—does it include only uninoculated compost? Statistical methods (ANOVA) are not fully justified—no mention of post-hoc tests or assumption checks.
Response 4: Agree. Thank you for your detailed review and the specific question about materials and methods. Replication details are showed in page 4, line 168. The bacterial inoculation process was done on page 3, lines 106-112. We have revised that CK was added in equal volumes of distilled water. Statistical methods (ANOVA) are revised for significant analyses of the same rows and columns in all tables, as are the tables in the Supplementary Material.
Comments 5: Results: Figures and tables need improvement for clarity: Figure 1 lacks clear differentiation between treatments. Plus the legend is not explained in the figure title. Statistical validation is incomplete: Confidence intervals (CIs) and effect sizes are missing. No mention of assumption checks for ANOVA (e.g., normality, homogeneity of variance).
Response 5: Thank you for pointing these out. Therefore, we have made these changes to the figures, please see page 6 of the revised manuscript, lines 216–223, and page 7, lines 252–253, and page 9, lines 287-286, which are shown in red. Furthermore, we considered that the addition of confidence intervals and effect sizes would complicate the figures , so we have hidden them. Thank you once again for the opportunity to enhance our work.
Comments 6: Discussion: Some conclusions are speculative and lack direct evidence:
The claim that Paenibacillus contributes to nitrogen fixation needs supporting enzymatic or genomic data. The role of temperature in microbial survival should be linked to composting efficiency. The study does not compare results with previous bacterial inoculation studies—how do findings align with existing research? Scalability for large-scale composting is not discussed—what are the practical limitations?
Response 6: Thank you for pointing these out. In the current study, we focused primarily on quantifying the efficiency of mycotoxin reduction efficiency and analyzing key physicochemical parameters during co-composting. Unfortunately, enzymatic and genomic were not performed due to limitations in analytical instrumentation and research scope. In future work, we plan to collaborate with laboratories equipped for enzymatic and genomic profiling to identify specific degradation pathways. However, some references could support the claim:
E.R. Oviedo-Ocaña, J. Soto-Paz, V.S. Torres, L.J. Castellanos-Suarez, D. Komilis, Effect of the addition of the Bacillus sp., Paenibacillus sp. bacterial strains on the co-composting of green and food waste, Journal of Environmental Chemical Engi-neering, 10 (2022). http://doi.org/10.1016/j.jece.2022.107816.
E.N. Grady, J. MacDonald, L. Liu, A. Richman, Z.C. Yuan, Current knowledge and perspectives of Paenibacillus: a review, Microbial Cell Factories, 15 (2016) 203. http://doi.org/10.1186/s12934-016-0603-7.
L.Y. Wang, J. Li, Q.X. Li, S.F. Chen, Paenibacillus beijingensis sp. nov., a nitrogen-fixing species isolated from wheat rhizosphere soil, Antonie Van Leeuwenhoek, 104 (2013) 675-683. http://doi.org/10.1007/s10482-013-9974-5.
This is the first study on the co-composting of moldy silage and rape straw, so we cannot compare results with previous bacterial inoculation studies. Moreover, the temperature differences in compost inoculated with each bacterial agent were not significant, and we did not correlate temperature with composting efficiency. Large-scale composting requires significant land area and specialized facilities, which may be resource-intensive in urban or space-constrained settings, and variability in feedstock composition (e.g., moisture, carbon-to-nitrogen ratio) can affect process stability at scale, necessitating rigorous preprocessing or blending strategies.
Comments 7: Figures and tables: Legends should specify statistical significance. Overlapping error bars make interpretation difficult. Supplementary tables should include raw mycotoxin degradation data over time.
Response 7: Thank you for pointing this out. Therefore, we have changed the legends to show statistical significance, please see page 6 of the revised manuscript, lines 216–223, and page 7, lines 252–253, page 9, lines 287-286, and page 10, lines 315-317, which are shown in red. We have done our best to separate the error bars and put the raw mycotoxin degradation data into the supplementary material. This adjustment contributes to clearer communication of the intended outcomes of our study's analysis. We have revised the appropriate parts of the text and the revised parts are shown in red.
Comments 8: Language and citation quality: Some sentences are too long and complex-restructure for clarity. Reference formatting is inconsistent (e.g., placeholder text in citations). Some statements lack citations—ensure all claims are properly referenced.
Response 8: Thank you for pointing this out. We extend our thanks for your meticulous review and for highlighting opportunities to refine our manuscript's clarity. We have changed some sentences more simplistically and their citations on page 6, lines 237-241, page 13, lines 375-377, page 18, lines 462-463, and page 19, lines 506-507, and revised the reference formatting on pages 20-23, lines 575-706. Your comments were instrumental in guiding these improvements, for which we are truly thankful. We have revised the appropriate parts of the text and the revised parts are shown in red.

Round 2
Reviewer 2 Report
Comments and Suggestions for Authors
I find that the authors have successfully addressed all reviewers' comments, resulting in significant improvements to the paper. Given its scientific rigor and the authors' approach, I recommend it for publication.